# Environmental Impact Assessment of an Ignition Pencil Coil by a Combination of Carbon Footprint and Environmental Priority Strategies Methodology

**Chih-Ming Chen** [1] and **Huey-Ling Chang** [2,*]

1   Department of Mechanical Engineering, National Chin-Yi University of Technology, Taichung 41170, Taiwan; cmchentc@gmail.com
2   Department of Chemical and Materials Engineering, National Chin-Yi University of Technology, Taichung 41170, Taiwan
*   Correspondence: hlchangtc@gmail.com; Tel.: +886-9-81583585

**Abstract:** This study presents a combined carbon footprint (CF) and environment damage assessment with a cradle-to-gate approach for an ignition coil. The process considers a data flow of product as the phases: raw materials preparation, part processing, final-product finishing, and packaging. The assessment was performed to explore an automotive ignition pencil coil during its developing phase. This study illustrated that a green product problem could be evaluated as a carbon footprint and environmental hazard. By using the conceptual flow to set up the assessment procedure, a product can be decomposed into several material ingredients to specify the input parameters in a Life Cycle Assessment. A total CF of an ignition coil can be investigated individually by each of the materials. The total CF of an ignition pencil coil equal to 0.5254 kgCO$_2$eq was calculated. The insulated filling and copper winding of an ignition coil generated the two most impacting processes in terms of CF (21.83% and 17.50%, respectively). EPS (Environmental Priority Strategies) methodology evaluates the environmental damage of the product in the product design process. As a result, the metal material has a seriously damaging impact on human health and inanimate resources, especially inanimate resources. The total CF generated by the newly devised ignition coil is over 39~62 percent less than a general type one that exists in the current market. The new ignition pencil coil also uses fewer raw materials and therefore reduces environmental damage to the Earth.

**Keywords:** carbon footprint; ignition coil; life cycle assessment; environmental priority strategies; environmental Impact

## 1. Introduction

As the growth of manufacturing activity increases, many industrial wastes are produced and discharged into our living environment. These industrial wastes can generate a variety of land and air pollution around the world. People have been paying attention to the environmental protection issues, such as land pollution, air pollution, climate change, and global warming, for several decades. In order for sustainable development on the Earth, it is time for all governments around the world to take steps to solve those environmental problems. The most effective method of measuring the environmental impact of the life cycle of a product is to quantify the greenhouse gas emissions required to produce it. It is known that CO$_2$ accounts for the largest proportion of greenhouse gases in the atmosphere. How to reduce CO$_2$ emission during the life cycle of a product to lessen the environmental impact is an important issue for industrial society. Then, how to reduce the usage of raw materials required for a product in a manufacturing process becomes the most popular issue for consideration. The objective was not to provide a comprehensive treatment of any single issue but highlight the types of issues that arise when carbon emission considerations are incorporated into design problems.

Matthews et al. [1] proposed that an estimation boundary of a carbon footprint is important. Their footprints should contain direct emissions, emissions from purchased energy, and supply chain emissions. Carbon emission can be divided into three categories, namely direct emissions, indirect emissions, and emissions in addition to the previous two kinds in the supply chain caused by other industries. Direct emissions of greenhouse gases arise from industry itself, which are the combustion of natural gas and oil during a manufacturing process. Indirect emissions of greenhouse gases are the carbon emissions generated from the energy purchase process. Direct emissions from industry are, on average, only 14% of the total supply chain carbon emissions, and direct emissions plus industry energy inputs are, on average, only 26% of the total supply chain emissions. [1] Many firms pursue an effective carbon moderation strategy to ensure their products meet environmental regulations.

Zhang et al. [2] reported that the product manufacturing stage is one of the main contributors to greenhouse gas (GHG) emissions. The manufacturing activity requires lots of equipment and raw materials to realize a modern mass production strategy. These machinery and raw materials within the manufacturing process will discharge GHG and trigger anthropogenic climate change.

It has been recognized that many manufacturing activities can result in potential environmental impacts. Mirasgedis et al. [3] performed an evaluation of the external cost attributable to the atmospheric pollution from the high environmental burden of industrial activities in the greater Athens area. The evaluation can be represented in monetary values to take into account human mortality and morbidity due to PM10 and $CO_2$ emissions, which are mostly discharged from the non-metallic minerals and oil processing industries. Benjaafar et al. [4] have claimed that the carbon emission parameters of a product could be associated with various decision-making variables for procurement, production, and inventory management. Manufacturers should understand the potential environmental impacts caused by each of their products. A methodology to meet this requirement is the Life Cycle Assessment (LCA). The LCA has been standardized for all types of products by the International Organization for Standardization and for electronics by the European Telecommunications Standards Institute (ETSI). The LCA method has been applied in a variety of products to present the environmental footprint for their life cycles, for example, Ibbotson and Kara [5], Duigou et al. [6].

Many researchers have examined the waste problem of consumer products from the point of view of greenhouse gas emission by the methodology of the LCA. An easily found product such as a polyethylene terephthalate (PET) bottle, for example. Shen et al. [7] studied the environmental impact of polyethylene terephthalate (PET) bottle-to-fiber recycling by using the methodology of the LCA. Four recycling cases, including mechanical recycling, semi-mechanical recycling, back-to-oligomer recycling, and back-to-monomer recycling, were considered. The result showed that PET bottle-to-fiber recycling offers important environmental benefits over single-use virgin PET fiber. Another visible problem can be found in China. For instance, abandoned television sets in China have become a serious environmental problem. Song et al. [8] pointed out that cathode ray tubes (CRT) and printed circuit boards (PCB) are those components which cause the most environmental damage from TV set manufacturing. As technology improves, old cathode ray tubes (CRT) and televisions give way to lighter weight, higher definition, flat-screen versions. According to Agilent, a CRT television requires about 0.3 Watt per square inch. According to CNET, high-end LED televisions use around 0.1 Watt per square inch at default settings and are closer to 0.075 Watt per square inch when calibrated to the individual user [9]. Min et al.'s [10] attempt to examine the consumers' preference or willingness to pay (WTP) a premium for eco-labeled products. This value amounts to 3.9% of the price of a conventional 43-inch LED TV and can be interpreted as the external benefit of an eco-labeled LED TV. People are willing to pay more for products with eco-labels. It shows that everyone is still willing to do their best for sustainable environmental protection. Tao et al. [11] developed a product life cycle cost (PLCC) model to support Taiwanese

light-emitting diode (LED) manufacturers in capacity planning for sustainable and resilient supply chain (SC) management. A notebook computer is widely produced every year in the world. Meyer and Katz [12] investigated the environmental impact of notebook computers by using the LCA for industries and policymakers to work together to develop sustainable products for the sake of climate change, and human and ecological health.

A model to assess the impact of global manufacturing on the embodied energy of products was presented by Kara et al. [13]. Six different products manufactured from various raw materials in a global manufacturing network were used to carry out the assessment. The results indicate that the embodied energy of products can be influenced by the manufacturing location, carriage weight, distance traveled, and transportation type used. Felic et al. [14] also demonstrated a multi-criteria decision-making model to compare the carbon footprint of electronic and electric equipment, which involves energy consumption derived from their manufacturing processes and their use phases to their end-of-life management. Elduque et al. [15] applied the methodology of the LCA to examine the environmental impact of the injection molding process. Aspects such as the infrastructure of the factory or waste treatment are part of the environmental impact of the injection molding process, but the most significant factor is electricity consumption. This environmental analysis has explored the processing of several parts made from high-density polyethylene, which have been characterized by measuring electricity consumption. To properly assess the actual environmental impact of a specific injection molding process, its real electricity consumption must be measured. Otherwise, the results would be quite far from the real values.

Climate change is a highly complex and challenging issue that makes government policymakers need access to objective information upon which to base their judgment about what substances might damage the environment before they take further action on climate change adaptation and mitigation strategies. Global warming potential (GWP) is a relative measure of how much heat is trapped by greenhouse gas in the atmosphere. The Intergovernmental Panel on Climate Change (IPCC) considers that greenhouse gas emissions caused by a product can be directly represented as a carbon dioxide equivalent for examining its potential impact on the environment.

Laurent et al. [16] have suggested that a carbon footprint can be an environmental impact indicator for materials based on the LCA approach. However, Andrae [17] believed that there are still many variabilities and inconsistencies in the LCA to study the carbon footprint of consumer electronics. Aspects of the LCA include raw material extraction, supplier transportation, manufacturing process, distribution, disposal transportation, and process. Morini et al. [18] have proposed using the LCA methodology and the CES Selector EcoAudit as a tool for the early stages of development and material selection in the design of new products.

Regarding these considerations, researchers have been thinking about how to reduce carbon emission and environmental damage during the life cycle of a product. The automobile has become an indispensable transportation method for mankind. However, abandoned automobiles also become a kind of indispensable waste. Product development is a set of activities beginning with the perception of a market opportunity and ending in the production, sale, and delivery of a product. Then, a design scheme should consider a devised object incorporated with environmental considerations. For example, Chang et al. [19,20] have investigated an overall carbon footprint of a general type of automobile ignition coil to understand the environmental impact caused by this product. The study showed that each production of an AS-944 ignition coil would generate an equivalent total of 1.394 kg carbon dioxide (kgCO$_2$eq). An AS-982 ignition coil will generate an equivalent total of 0.8694 kg carbon dioxide (kgCO$_2$eq). The assessment was operated by the SimaPro software.

Herrmann and Moltesen [21] observed differences appearing in their assessments, primarily originating from errors in the software databases for both inventory and impact assessments. The SimaPro and GaBi software are used by many users of the LCA worldwide

as a decision-support tool. It is clear in the interests of both software developers and LCA users that the observed differences have to be addressed, for example, through ring tests comparing the tools. The results of the analysis are representative of the differences obtained while using either one or the other, then the implication of the analysis is reliable.

The impact of human activities on the Earth's ecosystem is growing, bringing serious and problematic contradictions between the environment, the economy, and natural resources. Chen et al. [22,23] expounded quantitative modeling and simulations of regional ecology as the keys to realizing a strategy of regional sustainable development and developing an ecological footprint (EF) model for determining whether natural assets are over-utilized. This indicates that the burden of human activities on the natural environment is becoming increasingly serious. Urbanization has economic and social growth advantages, but at the same time, urbanization brings many problems such as air and environmental pollution. Karimian et al. [24] used various indicators to discuss economic development, environmental protection, and social welfare to explore sustainable urban development.

The analysis methods used in this study are carbon footprint and EPS 2000 (Environmental Priority Strategies) for carbon assessment and the environmental impact assessment. The ultimate purpose of the evaluation is to focus on improving the original design and process. Through the evaluation and analysis, the items with an environmental impact can be found, and an assessment of raw material requirements, design change proposals, or process improvements may be carried out. In this paper, carbon footprint and environmental damage are evaluated with a cradle-to-gate approach for a new ignition coil design. Results are presented and discussed in detail by life cycle phases in order to identify both environmental issues of the production process. The goal of this study is to produce a comprehensive view that considers green design problems by using carbon footprint as an indicator to provide a strategy aimed at improving the automobile part production process. An environmental friendliness assessment of a product in the design process can be investigated by environmental damage indicators. EPS 2000 (Environmental Priority Strategies) is applied to evaluate the environmental damage of a product in the product design process. The strategy can improve the chances of making the right choices in the selection of materials and processes, which could be useful in reducing the environmental impact of auto parts in the automobile industry.

## 2. Methodology and Materials

### 2.1. Life Cycle Assessment

According to ISO 14,040 (ISO, 2006 a) [25] and ISO 14,044 (ISO, 2006 b) [26], the implementation of the LCA can adopt the following phases: (1) Goal and scope definition, (2) Life cycle inventory, (3) Interpretation. The GWP 100a value of $CO_2$ is calculated by referring to 100 years from the time of the calculation to reflect the cumulative effects of the radiation of $CO_2$. For example, the GWP of methane was 25 in 2007, which means that one ton of methane after 100 years from the computing time, given that the cumulative effects of radiation are equivalent to the cumulative equivalent of 25 tons of $CO_2$ in the current time [27]. An IPCC GWP 100a methodology was applied in this study.

SimaPro software is a tool used for estimating GWP to help investigators calculate the carbon footprint of a product in the LCA. A procedure of factor characterization is needed to proceed at the beginning of the assessment. These characterized factors are the direct global warming potential caused by greenhouse gas emissions from an activity. A general principle is considered for the calculation procedure: (1) No need to consider any indirect effect; (2) Exclude the indirect emissions of nitrogen caused by the formation of nitrogen oxide; (3) Involve the carbon dioxide caused by the carbon monoxide emission; (4) Consider the offset of bio-absorption of carbon dioxide; (5) No need to consider the radiation effects of nitrogen oxides, water and sulfuric acid emissions arising from the troposphere to the stratosphere.

An overall emission of carbon dioxide equivalent referring to an activity, $CE_{CO_2}eq$ is defined as follows Equation (1):

$$CE_{CO_2}eq = D_j \times GWP_j = f_j \times \sum_i B_i^T \qquad (1)$$

where $GWP_j$ is the global warming potential of all greenhouse gases from the j-th activity, and $D_j$ refers to each amount of greenhouse gases; $f_j$ is an emission factor referring to the j-th greenhouse gas emission; $B_i^T$ is the information of activities in the manufacturing process.

A GWP value is usually calculated over a specific time interval, commonly by 20, 100, or 500 years. Most GWP evaluations are from the IPCC Fourth Assessment Report (AR4) published in 2007. The latest Synthesis Report (SYR) is the IPCC Fifth Assessment Report (AR5), released in 2014, which provides an overview of the state of knowledge concerning the science of climate change and emphasizes new results since the related information was updated. An exact definition of how GWP is calculated can be found in the IPCC AR4 (2007). For more information, please refer to the IPCC website (www.ipcc.ch/ accessed on 1 December 2021). A GWP is defined as the ratio of the time-integrated radioactive forcing capacity from the instantaneous release of 1 kg of a trace substance relative to that of 1 kg of a reference gas (IPCC, l990) [28]. The calculation Equation (2) is defined as the following:

$$\int_0^{TZ} a_y[y(t)]dt = GWP(y) \int_0^{TZ} a_g[g(t)]dt \qquad (2)$$

where $a_y$ is the radiating efficiency due to a unit increase in the atmospheric abundance of the substance (a metric unit in $Wm^{-2}\,kg^{-1}$); TZ is the time interval considered in the calculation. The two radiating efficiencies, $a_y$ and $a_g$, are not necessarily constant to time. Here, y(t) is the time-dependent decline in abundance of the instantaneous release of the substance at time t = 0, and g(t) is the corresponding quantities for the reference gas (i.e., $CO_2$)

*2.2. EPS 2000*

In the EPS impact assessment method, global average damage costs are estimated for emissions and resources, and the values of an average inhabitant are used [29]. The EPS (Environmental Priority Strategies) system is mainly aimed to be a tool for a company in the product development process. The evaluation model was developed by the Swedish Environmental Research Institute (IVL) in 1989. EPS 2000 is the 2000 version of the EPS system and it is a damage-oriented evaluation method [30,31]. It uses financial monetary units and willingness to pay (WIP) to represent investment and consumption that recover an object to the level of safety. The main objective of EPS 2000 is to serve as a tool for internal product development and improvement in the organization. The impact assessment by EPS 2000 on an environmental system is divided into 13 categories, and the damage assessment stage is divided into five environmental categories.

In the damage assessment stage, EPS 2000 takes the 13 environmental impact categories into five environmental protection categories, i.e., human health (including life expectancy, severe morbidity, morbidity, severe nuisance, and nuisance), ecosystem productivity (including crop production capacity, wood production capacity, fish and meat production capacity, soil acidification, the production capacity of irrigation water, and production capacity drinking of water), abiotic stock resource (depletion of reserves), biodiversity (including the extinction of species), culture and recreational value. However, item 5, cultural and tourist values, cannot be expressed by general indicators, so it is only a qualitative description and definition, not included in the assessment model. The detailed classifications will not be described here. A weighting indicator is imposed on the damage assessment. The step gives different weighting factors to the results of damage assessment according to the importance of different impacts. The "willing price" reflects the amount of social willingness to bear and "free from harm" indictaes the environmental impact of

products in five environmental protection categories. The unit of an aggregated indicator is ELU (Environmental Load Unit). After the three-step procedure: characterization, damage assessment, and weighting indicator process, the total impact in terms of money can be obtained by summing up the absolute value of each impact category. Through this stage, the impact of a product or process on the environment in the life cycle can be obtained. This is the so-called Single-score process. Through this assessment indicator, the decision-maker can analyze whether the product is environmentally friendly or not from the viewpoint of socioeconomic development. The WIP can be measured in terms of the different monetary systems of the different nations.

### 2.3. AS-507 Ignition Coil

A reduction in the total mass of an automobile can achieve better fuel economy to diminish greenhouse gas emissions to the environment. Any refinement in any single automotive part can greatly reduce the total mass of an automobile. An ignition coil is a small part belonging to an ignition system of an automotive engine. The ignition system is made of a battery, ignition switch, ignition coil, capacitor, distributor block, and plugs. As a part of the ignition system, the ignition coil is charged with the task of providing a spark plug with the required high voltage to generate an ignition spark between the center and earth electrode of the spark plug and ignite the air-fuel mixture. The electronic control unit allows current to flow through the ignition coil primary winding until it senses a triggering voltage pulse from the speed sensor or pickup coil. The function of an ignition coil is an electrical transformer that transforms 12 voltages of the battery to a maximum of 45 kilo-voltages during an ignition process. A downsized engine needs a smaller size ignition coil. For this purpose, a downsized and lightweight ignition coil is devised. The new ignition coil is defined as an AS-507 pencil ignition coil with specified dimensions of 71.1mm × 71.1mm × 180.3 mm and a mass of 231.5g. The sketch of this ignition coil is shown in Figure 1a,b. This ignition coil is made in a factory in Taiwan, so the data were collected from the factory in Taiwan. A factory in Taiwan collects data, then calculates and evaluates the damaging impact.

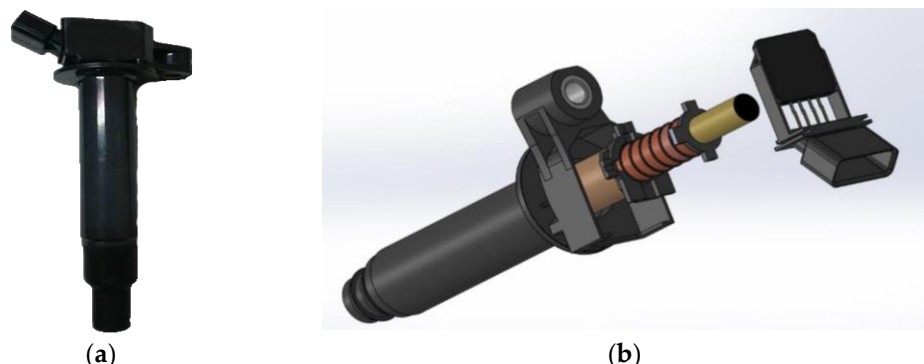

(**a**)    (**b**)

**Figure 1.** (**a**) A solid image of AS-507 pencil coil, (**b**) 3-D model of AS-507 pencil coil.

The AS-507 pencil ignition coil can be applied to various automobiles as a fitment by car makers (Some applicable models and numbers are shown in Table 1). Please refer to the website http://www.asia-traffic.com.tw (accessed on 1 December 2021) for more details. This new automobile part is compatible with 138 vehicles. Please refer to the website http://www.ebay.com (accessed on 1 December 2021) for more information.

**Table 1.** Pencil ignition coil (AS-507) applied as an automobile fitment by car maker.

| Standard Motor Products | Original Number | Corresponding to the Car Maker | Corresponding Model | Corresponding Vehicle Type |
|---|---|---|---|---|
| UF-333 UF-494 | 90919-02243 | Toyota | Sedan, Sport Utility Vehicle | Camry Highlander RAV4 Solara |
| | 90919-02244 | Toyota | Sedan, Sport Utility Vehicle | Camry Corolla Highlander Matrix RAV4 Solara |
| | | Scion (Division of Toyota) | Sedan | tC xB |
| | 90080-19023 | Toyota | Sedan | Camry Solara |
| | 90919-02266 | Toyota | Sedan, Sport Utility Vehicle | Camry Corolla Highlander Matrix RAV4 Corolla |
| | 90919-19023 | Scion (Division of Toyota) | Sedan | tC xB |
| | | Lexus | Sedan | HS250h |
| | | Toyota | Sedan, Sport Utility Vehicle | Matrix RAV4 Highlander Solara Camry |
| | | Scion (Division of Toyota) | Sedan | tC |

### 2.4. System Boundaries and Material Composition

An automotive ignition coil is investigated from raw materials preparation to a final product in this study. The IPCC GWP 100a model is applied to calculate the potential global greenhouse effect of the associated raw materials by converting them into $CO_2$ equivalent. The overall CF of an ignition coil can be observed by a cradle-to-gate approach. The GWP calculation is based on the IPCC GWP 100a method to explore how a potential global greenhouse effect is converted into $CO_2$ equivalent. The overall CF of an AS-507 ignition coil from raw materials to a final product can be examined. The inquiry of the related CF data can be obtained from the inventory specifications, field inventory, and surveyed literature. The information on electrical energy consumption in production was recorded by a specified factory in Taiwan. Data on the consumption of raw materials were mainly collected and computed by the factory which manufactured the product. The materials of an AS-507 pencil ignition coil, the material composition of the component parts, and the weighing quality are shown in Table 2. Due to a lack of direct resources, the greenhouse gas analysis of the raw materials was performed according to the related literature and database.

The 7.3 version of software SimaPro is used in this study. The software can produce a tree diagram to clearly show the environmental impact as the tree diagram clearly exhibits the branch of the respective input energy and material. In the tree-branch subsystem, the system represents the data in a measurable way, based on an expression similar to a thermometer, and quickly determines the impact of the materials and the energy consumption

on the environment. Therefore, instead of performing an assessment for a specific system, the analysis scope is focused on the manufacturing of an ignition pencil coil (type AS-507). The system boundary of the product is specified by a cradle-to-gate method. According to the production process and the order of adding each raw material, draw the system boundary as shown in Figure 2. The choice of the functional unit of GHG emissions has important implications for the interpretation of the results. In most cases, GHG emission is defined as a total of $CO_2$eq per kg of product or a total of $CO_2$eq per kg of the production system. In this study, the comparison basis across products is selected as per "ignition coil," thus, the system boundaries are as shown in Figure 2.

**Table 2.** Material of an AS-507 pencil ignition coil.

| Component | Packaging Materials | Material Composition | Mass (g) |
|---|---|---|---|
| Cushion | | Silicone | 0.0491 |
| Primary spools | | PPS (Polyphenylene sulfide) | 7.0000 |
| Secondary spools | | PPS | 5.5000 |
| Insulating body | | PBT (Polybutylene terephthalate) | 17.0000 |
| Insulating resin | | Epoxy resin | 42.1000 |
| Jacket | | Silicone | 13.8000 |
| Terminal | | Copper | 0.3000 |
| Coil | | Copper | 47.6000 |
| Secondary terminal | | Copper | 0.2000 |
| Terminal block | | Copper | 0.7000 |
| Core | | Iron | 37.6000 |
| Bush | | Iron | 3.0000 |
| High-voltage terminal | | Aluminum | 0.5000 |
| Spring | | Stainless steel | 0.2000 |
| Tin wire | | Tin | 0.2000 |
| | Tape | PP (Polypropylene) | 0.0067 |
| | Carton | paper | 52.9000 |
| | plastic bags | PE (Polyethylene) | 4.7000 |

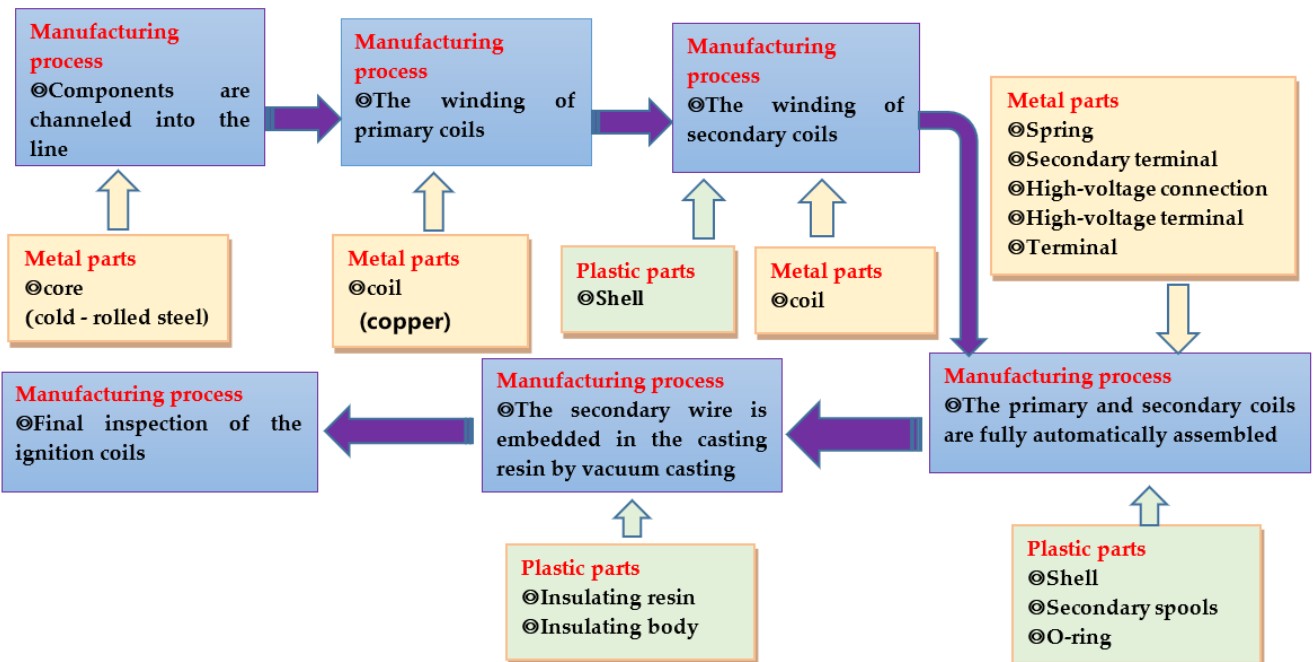

**Figure 2.** System boundaries of an AS-507 ignition coil.

## 3. Results and Discussion

### 3.1. Carbon Footprint

Based on the guidance of this industry, the carbon footprints were performed per the so-called definition of "one unit of the ignition coil." The elements of an ignition coil are divided into three parts, i.e., polymer materials, metal materials, and packaging materials. CF calculates the amount of greenhouse gas emissions caused by a particular activity or entity and measures kilograms of carbon dioxide equivalent ($CO_2$eq). The net-diagram analysis was illustrated in Figures 3–5. The carbon footprint of a different element was calculated and listed in Table 3.

As shown in Figure 3, the categories of polymer material are insulating resin, insulating body, secondary spools, primary spools, cushion, and jacket, and their corresponding GHG emissions are 0.1147, 0.0819, 0.0305, 0.0389, 0.0002, and 0.0373, all measured in kg$CO_2$eq, respectively. For the category of metal materials, as shown in Figure 4, the largest portion of the carbon emission sources is from the coil (0.0897 kg$CO_2$eq), followed by the core (0.0338 kg$CO_2$eq). In the category of package materials, as shown in Figure 5, cartons, plastic bags, and tape have corresponding carbon footprints of 0.0099, 0.0780, and 1.322 $\times 10^{-5}$ kg$CO_2$eq, respectively. Here, it should be noted that the quantity of 1.322 $\times 10^{-5}$ kg$CO_2$eq emitted from tape should be actually neglected and denoted as 0.0000 in this case. However, the tape is usually used during the packaging process and is made of polypropylene which can also produce pollution to the environment. This expression is only for your best knowledge to make people carefully notice the potential pollution caused by this material.

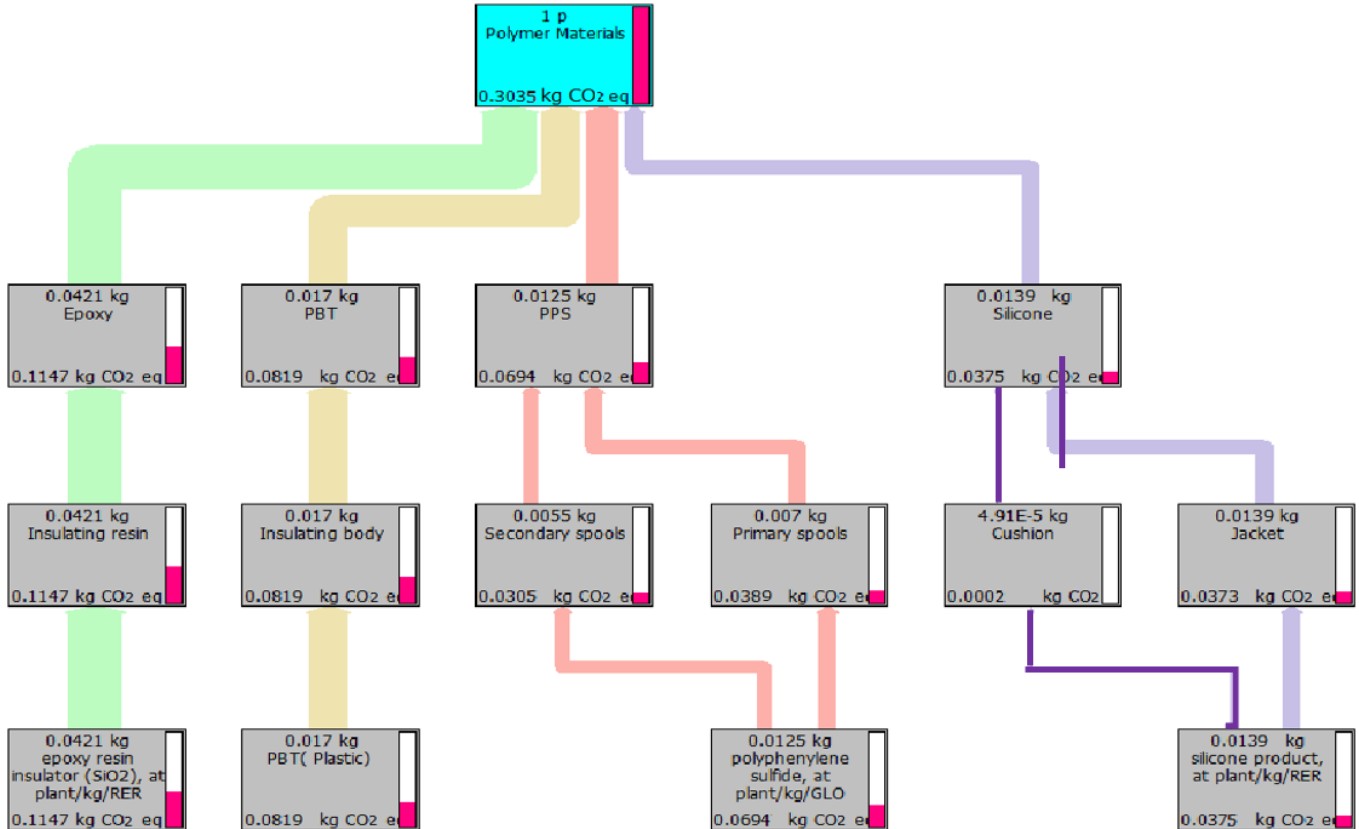

**Figure 3.** Net-analysis diagram of the polymer materials by IPCC GWP.

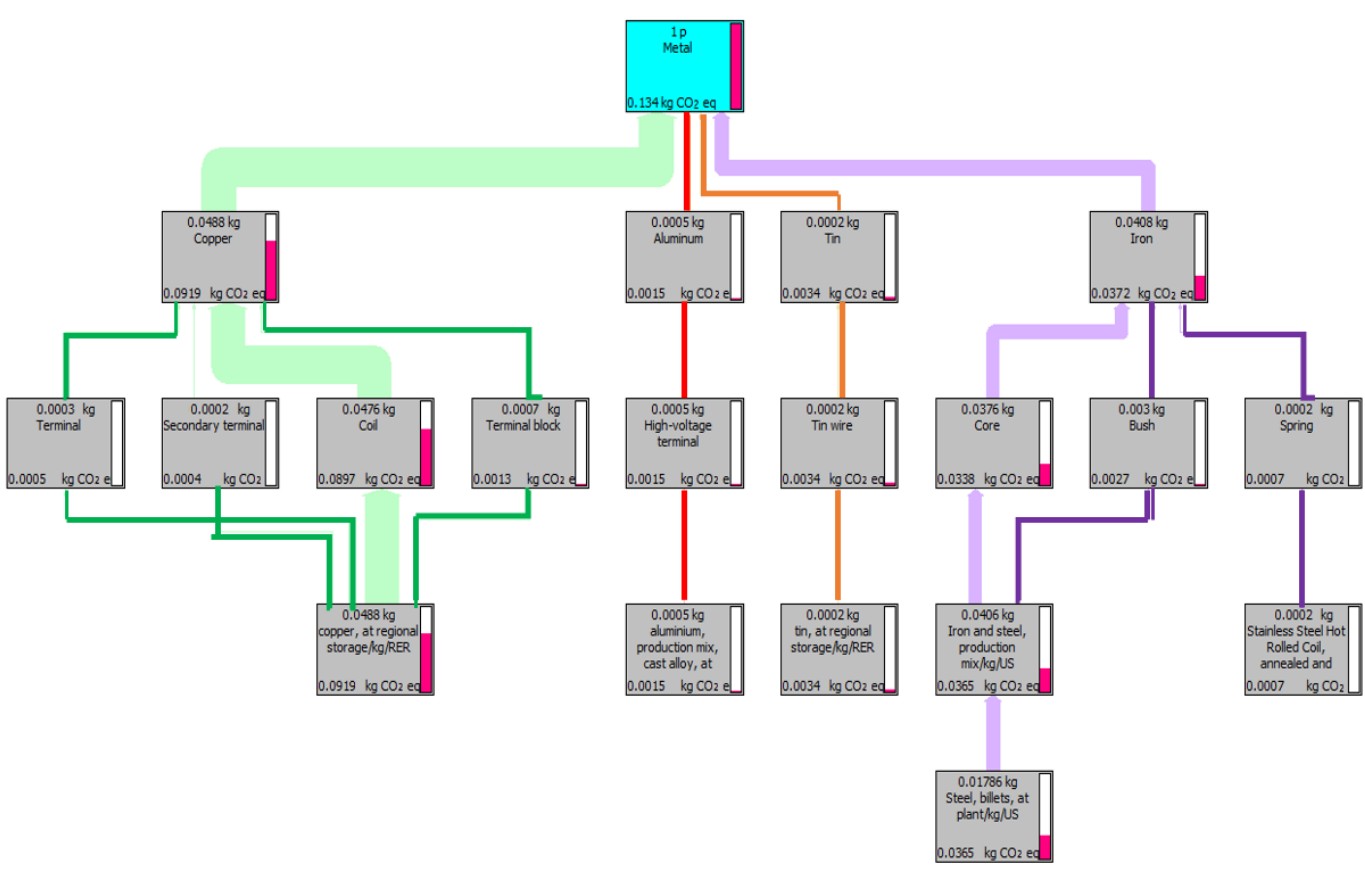

**Figure 4.** Net-analysis diagram of the metal materials by IPCC GWP.

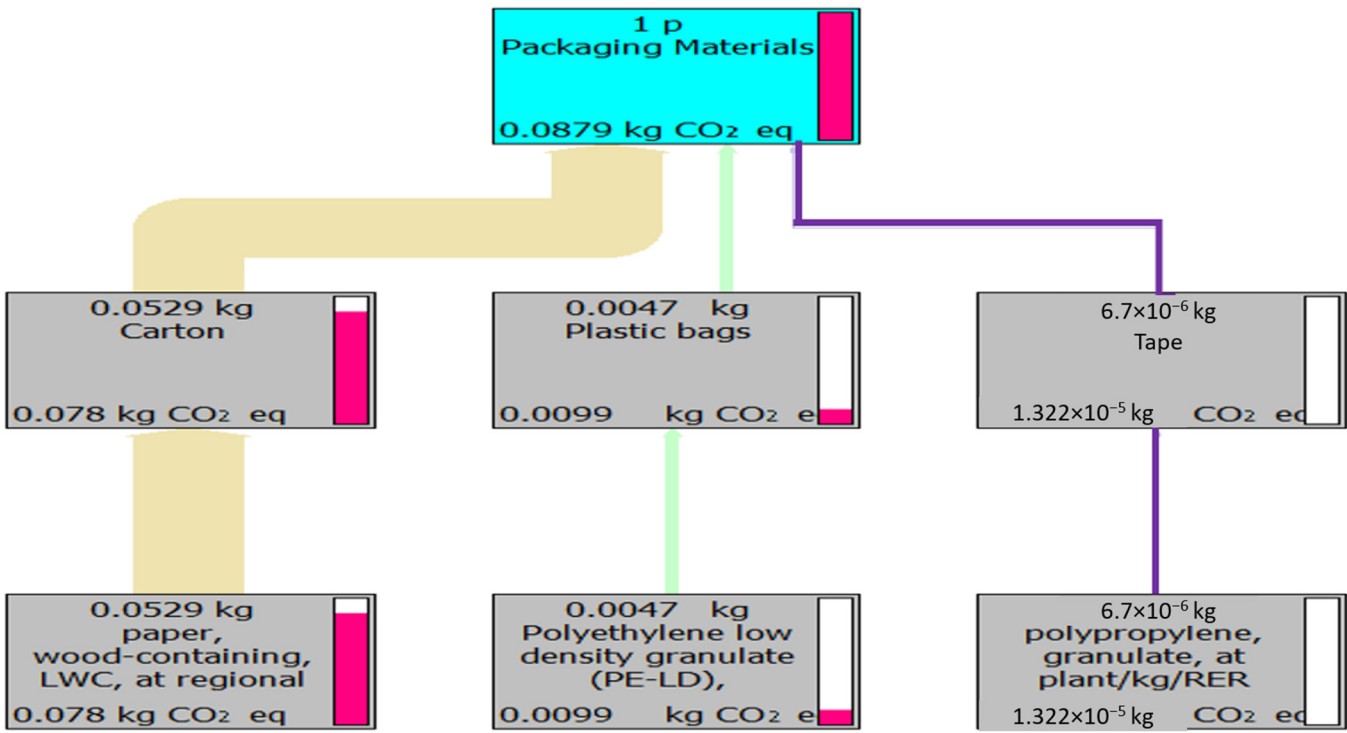

**Figure 5.** Net-analysis diagram of the packaging materials by IPCC GWP.

**Table 3.** CF generated by a pencil ignition coil (AS-507).

| Emission Source | Total Mass Usage (g) | Carbon Footprint $(KgCO_2eq)$ | Emission Source Carbon Footprint (%) |
|---|---|---|---|
| Epoxy resin | 42.1000 | 0.1147 | 21.83 |
| PBT | 17.0000 | 0.0819 | 15.60 |
| PBS | 12.5000 | 0.0694 | 13.20 |
| Silicone | 13.9000 | 0.0375 | 7.13 |
| Copper | 48.8000 | 0.0919 | 17.50 |
| Iron | 40.8000 | 0.0372 | 7.07 |
| Tin | 0.2000 | 0.0034 | 0.65 |
| Aluminum | 0.5000 | 0.0015 | 0.30 |
| Paper | 52.9000 | 0.0780 | 14.84 |
| PE | 4.7000 | 0.0099 | 1.88 |
| PP | 0.0067 | 0.0000 ($1.322 \times 10^{-5}$) | 0.00 |
| Total | 233.4067 | 0.5254 | 100.00 |

From the analysis, as seen in Table 3, each production of an AS-507 ignition coil generates a total of 0.525 kgCO$_2$eq. The production of the required materials for an ignition coil is a necessary process that is an unavoidable contributor to carbon emissions. The insulating epoxy resin is the most carbon-emitting contributor in the category of raw materials, producing a total of 0.1147 kgCO$_2$eq (or 21.83% of the total CF). The production of the coil and insulating outer case of the ignition coil accounts for the second and third largest contribution to greenhouse gas emissions, equivalent to 0.0919 kgCO$_2$eq (or 17.50% of the total CF) and 0.0819 kgCO$_2$eq (or 15.60% of the total CF), respectively. Mostly, epoxy resin is used for the insulating filling of the ignition coil for better performance and heat dissipation. Carton is one of the packaging materials in this activity. It presents a relatively large component in this category that generates carbon emissions up to 0.0780 kgCO$_2$eq (or 14.84% of the total CF). A number of CFs cannot be ignored in this situation. That the carbon emission of epoxy resin for the ignition coil is the largest, followed by the copper coil, and the third is the insulating body. The insulation function is very important in the high voltage environment, and the coil enhancement voltage cannot be replaced by the ignition system. Involving performance requires manufacturers to consider vehicle design. For example, we also study changes in epoxy resin composition to enhance performance [32], which are all directions for consideration. However, the paper part of the packaging material occupies four places in the data. Obviously, there should be an improvement in changing packaging materials. Reducing the use of material and pollutants in finished products is also a way to be environmentally sustainable. Manufacturing changes to the overall ignition coil style can significantly improve results.

If the amount of each material in the finished product is reduced while still achieving function, this would be a good way to reduce the waste of resources. Therefore, we study and compare the carbon emissions and economic impact data of the three ignition coils in the same analysis model and provide the engineering department with new considerations for the production process. Ignition coils of different shapes and material composition ratios and the results were used to compare with the total CF of a general type ignition coil described in Chang's study [19,20]. A general AS-944 ignition coil generates a total of 1.394 kgCO$_2$eq, and an AS-982 ignition coil generates a total of 0.8694 kgCO$_2$eq (i.e., polymers, metals, and packaging materials contribute 0.4543, 0.3227, and 0.0879 kgCO$_2$eq, respectively).

By comparing the CFs of three types of automotive ignition coils, it is found that a downsized and lightweight pencil ignition coil (AS-507) can reduce over 62 percent of a total CF than a general ignition coil (AS-944) and reduce over 39 percent of a total CF than an ignition coil (AS-982).

### 3.2. Environment DAMAGE Assessment

The unit index of EPS 2000 is ELU (Environmental Load Unit). Its evaluation process is Characterization, Damage Assessment, Weighting, and Single-Score. The purpose of this method is to measure the environmental friendliness of products in the design process from the economic point of view of the willingness to pay price. Tables 4 and 5 show the damage assessment and weighting processes on the categories of materials and electricity consumption for an AS-507 ignition pencil coil by EPS 2000. From Table 4, the damage assessment of an AS-507 ignition coil by EPS2000, the recovery of the abiotic stock resource will cost the most money; Human health is second. Table 5 shows the weighting analysis and adds a percentage. Through the weighting process, we determine which has the greatest impact. In this step, the relative importance of each category of impact is determined. The abiotic stock resource has the greatest impact. The manufacture of the AS-507 ignition pencil coil company generates an important impact on abiotic resources. The results show that 98.29% of the abiotic stock resource damage comes from metal. In the category of human health, as shown in Table 5 are metal, polymer, and energy consumption which corresponds to the weighing percentage of 67.65%, 20.80%, and 11.55%, respectively. In the category of ecosystem production capacity, there are polymer, metal, and energy consumption which have corresponding weighing percentages of 41.0%, 38.7%, and 20.3%, respectively.

**Table 4.** Damage assessment on AS-507 pencil ignition coil by EPS2000.

| Impact Category | Unit | Total | Polymer | Metal | Energy Consumption |
|---|---|---|---|---|---|
| Human health | ELU | $2.38 \times 10^{-1}$ | $4.95 \times 10^{-2}$ | $1.61 \times 10^{-1}$ | $2.75 \times 10^{-2}$ |
| Ecosystem production capacity | ELU | $1.906 \times 10^{-3}$ | $7.82 \times 10^{-4}$ | $7.38 \times 10^{-4}$ | $3.86 \times 10^{-4}$ |
| Abiotic stock resource | ELU | 8.80 | $1.21 \times 10^{-1}$ | 8.65 | $2.88 \times 10^{-2}$ |
| Biodiversity | ELU | $1.08 \times 10^{-3}$ | $4.27 \times 10^{-4}$ | $3.89 \times 10^{-4}$ | $2.61 \times 10^{-4}$ |

**Table 5.** Weighting process on AS-507 pencil ignition coil by EPS 2000.

| Impact Category | Unit | Total | Polymer | Metal | Energy Consumption |
|---|---|---|---|---|---|
| Human health | Pt. | $2.38 \times 10^{-1}$ (100%) | $4.95 \times 10^{-2}$ (20.80%) | $1.61 \times 10^{-1}$ (67.65%) | $2.75 \times 10^{-2}$ (11.55%) |
| Ecosystem production capacity | Pt. | $1.906 \times 10^{-3}$ (100%) | $7.82 \times 10^{-4}$ (41.0%) | $7.38 \times 10^{-4}$ (38.7%) | $3.86 \times 10^{-4}$ (20.3%) |
| Abiotic stock resource | Pt. | 8.80 (100%) | $1.21 \times 10^{-1}$ (1.38%) | 8.65 (98.29%) | $2.88 \times 10^{-2}$ (0.33%) |
| Biodiversity | Pt. | $1.08 \times 10^{-3}$ (100%) | $4.27 \times 10^{-4}$ (39.81%) | $3.89 \times 10^{-4}$ (36.02%) | $2.61 \times 10^{-4}$ (24.17%) |

Thus, the result after the Single-score process on the materials and electricity consumption of the AS-507 ignition coil can be obtained as shown in Table 6. Polymer materials and energy are the two most serious causes of environmental damage in this case. Polymer material has a more serious impact on ecosystem production capacity and biodiversity than the others. Metal material has the most damage impact on human health and abiotic stock resource, especially 98.29% damage to the abiotic stock resource. Resource consumption is usually the highest economic cost in the life cycle of the product. In general, the metal and polymer material of an ignition coil is harder to recycle than the other products, so the environmental impact is relatively considerable. The environmental impact of electricity consumption is dependent on what kind of power system is provided in the local area. The environmental impact of electricity consumption is relatively significant in this study. A given high score for energy consumption in an assessment means that the power system should consider more green power suppliers for local governments in the future.

**Table 6.** Single-score process on AS-507 pencil ignition coil by EPS 2000.

| Material | Unit | Total | Human Health | Ecosystem Production Capacity | Abiotic Stock Resource | Biodiversity |
|---|---|---|---|---|---|---|
| Polymer | Pt. | $6.23 \times 10^{-2}$ | $4.95 \times 10^{-2}$ | $7.82 \times 10^{-4}$ | $1.21 \times 10^{-2}$ | $4.27 \times 10^{-4}$ |
| Metal | Pt. | $8.81 \times 10^{-3}$ | $1.61 \times 10^{-1}$ | $7.38 \times 10^{-4}$ | 8.65 | $3.89 \times 10^{-4}$ |
| Energy consumption | Pt. | $5.68 \times 10^{-2}$ | $2.75 \times 10^{-2}$ | $3.86 \times 10^{-4}$ | $2.88 \times 10^{-2}$ | $2.61 \times 10^{-4}$ |

## 4. Conclusions

This paper presents a combined CF and environment damage assessment to demonstrate that a new design of ignition coil has less environmental impact than the other general types. That a downsized and light-weight ignition coil (product type AS-507) can reduce over 62% of CF than a general ignition coil (product type AS-944). The EPS 2000 model results show that polymer material has more serious damage to ecosystem production capacity and biodiversity. Metal materials cause more severe damage to human health and abiotic stock resource, especially 98.29% damage to the abiotic stock resource. This assessment model provides automobile manufacturers with great information on a selection of the required automobile parts in their assembly lines.

The changes in epoxy resin composition can enhance performance. We also conducted research on material composition changes and performance optimization, combined with analysis methods, hoping to give designers more improvements. Different carbon emissions can give designers an understanding of the impact of items on the environment and can also enable designers to think and follow directions for new designs of ignition coils in the future. Quantitative assessment of environmental damage costs and repairing environmental impacts can help businesses and consumers to operate more friendly and sustainable. The above conclusions can provide relevant industries as a reference or basis for analysis.

**Author Contributions:** Formal analysis, C.-M.C. and H.-L.C.; Investigation, C.-M.C. and H.-L.C.; Methodology, H.-L.C.; Project administration, H.-L.C. All authors have read and agreed to the published version of the manuscript.

**Funding:** This research received no external funding.

**Institutional Review Board Statement:** Not applicable.

**Informed Consent Statement:** Not applicable.

**Data Availability Statement:** Not applicable.

**Acknowledgments:** The research was supported by Ministry of Science and Technology of Taiwan. The researchers hereby express the gratitude.

**Conflicts of Interest:** The authors declare no conflict of interest.

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
