# Peer review of "Environmental Impact Assessment of an Ignition Pencil Coil by a Combination of Carbon Footprint and Environmental Priority Strategies Methodology"

_sustainability, doi:10.3390/su14084783_

Round 1
Reviewer 1 Report
Some of the mentioned probelms statements are rather out of dated, such as CRT is no longer a problem in the recycling centres whilst LED/LCD TV sets and big screen mobile phones have already flooded in as new types of electric wastes.
It is not clear if the proposed solution can be extended to any other applications.
Author Response
Some of the mentioned probelms statements are rather out of dated, such as CRT is no longer a problem in the recycling centres whilst LED/LCD TV sets and big screen mobile phones have already flooded in as new types of electric wastes.
Answer: Thanks for the suggestion. Added new information, described on line 87-99.
It is not clear if the proposed solution can be extended to any other applications.
Answer: This analysis method can provide a basic assessment of the environmental impact of products.Detail described on line 488-494.

Reviewer 2 Report
The article entitled, “Environmental Impact Assessment of an Ignition Pencil Coil 2 by a Combination of Carbon Footprint and Environmental Priority Strategies methodology” is a well-documented research article presenting a combined carbon footprint (CF) and environment damage assessment, with a cradle to gate approach, for an ignition coil. The authors want to describe the process considers a data flow of product as the phases: raw materials preparation, part processing, final-product finishing, and packaging. But this paper has some major shortfalls.
While revision of the article following points must be considered:
The introduction portion is very long. It must be shortened by describing only relevant points.
Methodology and materials method portion should be merged.
The table 1(Table 1. Pencil ignition coil (AS-507) applied as an automobile fitment by car makers) should be in the supplementary file mentioning in the main text.
The section 3.2 System boundaries and material composition has no proper references. It must be re-written with the proper citation.
The Table 2. Material of an AS-507 pencil ignition coil, not mentioned anywhere in the text.
The Figure 3. Net-analysis diagram of the polymer materials by IPCC GWP, is not easily readable. Please make it more prominent and clearer. Same for Fig. 4 and 5.
The manuscript has no proper discussion. The authors failed to describe the science or logic behind there findings. After each results a proper discussion with suitable reference must be included for justification of their findings. The entire results and discussion portion must be re-farmed accordingly.
The conclusion portion must be shortened with proper ‘take-home’ messages or relevant findings.
Author Response
The introduction portion is very long. It must be shortened by describing only relevant points.
Answer: Thanks for the suggestion. Revised Introduction.
Methodology and materials method portion should be merged.
Answer: Thanks for the suggestion. Methodology and materials were merged in Section 2.
The table 1(Table 1. Pencil ignition coil (AS-507) applied as an automobile fitment by car makers) should be in the supplementary file mentioning in the main text.
Answer: Thanks for the suggestion. There described on line 283-284.
The section 3.2 System boundaries and material composition has no proper references. It must be re-written with the proper citation.
Answer: Thanks for the suggestion. There were rewritten in section 2.4 and line 310-318.
The Table 2. Material of an AS-507 pencil ignition coil, not mentioned anywhere in the text.
Answer: Thanks. There described on line 299-301.
The Figure 3. Net-analysis diagram of the polymer materials by IPCC GWP, is not easily readable. Please make it more prominent and clearer. Same for Fig. 4 and 5.
Answer: Thanks for the suggestion. The images have been enhanced prominent and clearer.
The manuscript has no proper discussion. The authors failed to describe the science or logic behind there findings. After each results a proper discussion with suitable reference must be included for justification of their findings. The entire results and discussion portion must be re-farmed accordingly.
Answer: Thanks for the suggestion. There were added more discussed and new information, described on line 392-407.
The conclusion portion must be shortened with proper ‘take-home’ messages or relevant findings.
Answer: Thanks for the suggestion. The shortened conclusion were described on line 477-480.

Reviewer 3 Report
The study by Chen et.al investigated the environmental impact of AS-507 ignition coil by combination of carbon foot print and environmental priority strategies. There are several problems in this manuscript. Therefore, I would rather to suggest major revision and the following comments must be addressed before any decision for publication.
1- Please clearly mention the novelty of your study in the last paragraph of the introduction as well as briefly in your abstract.
2- Based on MS content, it’s necessary to alter the title to fit the content.
3- The English of the paper need much more effort to be understandable by readers.
4- Line_11: Please be consistent in using cradle-to-gate
5- Line_11: this line is inconsistent with your title. Which method was used finally?
6- In introduction, the author must add some nice introduction about different ignition pencil coil.
7- In introduction, expand the emphasis on the subject matter by focusing on recent articles in the Sustainability. Authors must modify introduction by stating the following points "Problems, Possible solution, Disadvantages of these solutions, Author's idea, Advantages of authors idea, etc." Introduction in overview form is not recommended. Also to widen your literature review, the following references can be cited:
doi:10.3390/su11051353
doi:10.4028/www.scientific.net/AMM.253-255.244
https://doi.org/10.5194/isprs-annals-IV-4-W2-153-2017
https://doi.org/10.1016/j.jwpe.2022.102696
8- Line 45: please explain more about the third category which is the combination of the first two emissions. Bring an example may make this clearer.
9- Line 52-55: please revise it. Not clear at all.
10- In Section 2.1, the first paragraph is more like the research background. Please move it to introduction part.
11- Equation 1 and 2 are not aligned.
12- In Section 2.2, the allocation of the first and second paragraphs is unreasonable. The five environmental categories mentioned at the end of the first paragraph are the same as those described at the beginning of the second paragraph.
13- In Section 2.2, the environmental damage assessment stage of EPS 2000 is divided into five categories, which can be displayed in the form of tables or pictures, which is more intuitive and lacks the form of tables or pictures.
14- Section 3.1: why you select this ignition coil to study about? The authors did not mention about the manufacturer of this device.
15- The first paragraph of Section 3.2 talks about the experimental methods and steps, which has nothing to do with the title of Section 3.2 'system boundaries and material composition'.
16- Section 3.2, table 2 does not have any paragraphs to quote and describe. It is recommended to describe Table 2.
17- Section 4.1, the first paragraph refers to geographical analysis. Please explain which part reflects geographical analysis?
18- In Section 4.1, the analysis of Table 3 in the third paragraph should be placed after the first paragraph, which is not conducive to readers looking at the table for comparison while reading.
19- In Section 4.1, the fifth paragraph still talks about the results of the fourth paragraph, which should be placed in the fourth paragraph.
20- In Section 4.1, need to discuss more about the results of CF.
21- Please revise the conclusion part. The conclusion part lacks a summary description and is more a restatement of the results.
22- The conclusion part can add future prospects or some suggestions and directions.

Author Response
1.Please clearly mention the novelty of your study in the last paragraph of the introduction as well as briefly in your abstract.
Answer: Thanks for the suggestion. There were described on line 167-172.
2.Based on MS content, it’s necessary to alter the title to fit the content.
Answer: Thanks. There were rewritten content.
3.The English of the paper need much more effort to be understandable by readers.
Answer: Thanks. There were improved.
4.Line_11: Please be consistent in using cradle-to-gate
Answer: Thanks for the suggestion. Changed in using cradle-to-gate (on line 11).
5.Line_11: this line is inconsistent with your title. Which method was used finally?
Answer: Thanks. There were described on line 167-172.
6.In introduction, the author must add some nice introduction about different ignition pencil coil.
Answer: Thanks. There were described on line 403-407, and references 19-20.
7.In introduction, expand the emphasis on the subject matter by focusing on recent articles in the Sustainability. Authors must modify introduction by stating the following points "Problems, Possible solution, Disadvantages of these solutions, Author's idea, Advantages of authors idea, etc."…
Answer: Thanks for the suggestion. The introduction were rewritten on line 156-166.
8.Line 45: please explain more about the third category which is the combination of the first two emissions. Bring an example may make this clearer.
Answer: Thanks for the suggestion. The was rewritten on line 51-53.
9.Line 52-55: please revise it. Not clear at all.
Answer: Thanks for the suggestion. The was rewritten on line 44-46.
10.In Section 2.1, the first paragraph is more like the research background. Please move it to introduction part.
Answer: Thanks for the suggestion. They were moved to introduction part.
11.Equation 1 and 2 are not aligned.
Answer: Thanks. Arranged aligned.
12.In Section 2.2, the allocation of the first and second paragraphs is unreasonable. The five environmental categories mentioned at the end of the first paragraph are the same as those described at the beginning of the second paragraph.
Answer: Thanks. The was rewritten on line 230-231.
13.In Section 2.2, the environmental damage assessment stage of EPS 2000 is divided into five categories, which can be displayed in the form of tables or pictures, which is more intuitive and lacks the form of tables or pictures.
Answer: Thanks. They was described on line 242-246.
14.Section 3.1: why you select this ignition coil to study about? The authors did not mention about the manufacturer of this device.
Answer: They was described on line 275-279.
15.The first paragraph of Section 3.2 talks about the experimental methods and steps, which has nothing to do with the title of Section 3.2 'system boundaries and material composition'.
Answer: Thanks. The revised title was incorporated in the Section 2.
16.Section 3.2, table 2 does not have any paragraphs to quote and describe. It is recommended to describe Table 2.
Answer: Thanks. There described on line 299-301.
17.Section 4.1, the first paragraph refers to geographical analysis. Please explain which part reflects geographical analysis?
Answer: Thanks. This ignition coil is made in Taiwan factory, so the data is collected in Taiwan factory. There described on line 346-348.
18.In Section 4.1, the analysis of Table 3 in the third paragraph should be placed after the first paragraph, which is not conducive to readers looking at the table for comparison while reading.
Answer: Thanks. Position is adjusted.
19.In Section 4.1, the fifth paragraph still talks about the results of the fourth paragraph, which should be placed in the fourth paragraph.
Answer: Thanks. Results discuss adding and adjusting positions.
20.In Section 4.1, need to discuss more about the results of CF.
Answer: Thanks. Results discuss adding in line 392-411.
21.Please revise the conclusion part. The conclusion part lacks a summary description and is more a restatement of the results.
Answer: Thanks for the suggestion. The more conclusion were described on line 477-480.
22.The conclusion part can add future prospects or some suggestions and directions.
Answer: Thanks for the suggestion. The more suggestion were described on line 481-494.

Round 2
Reviewer 2 Report
The manuscript is now significantly improved. But still some correction is needed. The quality of Fig. 3, 4 and 5 still need improvement for better understanding. I am still not satisfied with the discussion portion of the section "Environment damage assessment". Please improve this section. Conclusion portion is still very big. Short this portion bullet points.
Author Response
The manuscript is now significantly improved. But still some correction is needed. The quality of Fig. 3, 4 and 5 still need improvement for better understanding. I am still not satisfied with the discussion portion of the section "Environment damage assessment". Please improve this section. Conclusion portion is still very big. Short this portion bullet points.
Answer: Thanks for your suggestions and corrections to make this manuscript more complete. The Figures have improved. "Environment damage assessment" add description in line 422-425. The conclusions were rewritten in two paragraph.

Reviewer 3 Report
Thank you for your nice effort in revising the manuscript. However, there are still some errors in your manuscript that need to be modified.
- The English of the paper is my main concern. Specially those parts added after revision seems written carelessly and must undergone extensive English proof.
- The conclusion should be revised and written in one or maximum two paragraph.
- L 345: this sentence can be removed, as the authors did nothing about geographic analysis.
- L 346-348: This should be moved to the somewhere in section 2.3
- L 488-495: not clear at all.

Author Response
Thank you for your nice effort in revising the manuscript. However, there are still some errors in your manuscript that need to be modified.
The English of the paper is my main concern. Specially those parts added after revision seems written carelessly and must undergone extensive English proof.
The conclusion should be revised and written in one or maximum two paragraph.
Answer: Thanks for your suggestions and corrections to make this manuscript more complete. The conclusions were rewritten in two paragraph. It has been improved.
L 345: this sentence can be removed, as the authors did nothing about geographic analysis.
Answer: Thanks for your suggestion, it has been removed.
L 346-348: This should be moved to the somewhere in section 2.3
Answer: Thanks for your suggestion, it has been removed to section 2.3. Line 280-282.
L 488-495: not clear at all.
Answer: Thanks very much. This were rewritten in line 472-473, and briefly explained in the conclusion.

Round 3
Reviewer 3 Report
Thank you for your efforts. Although the English of the paper has still some problems, but the paper can now be considered for publication.
Author Response
Comments and Suggestions for Authors
Thank you for your efforts. Although the English of the paper has still some problems, but the paper can now be considered for publication.
Answer: Thanks for your suggestions. Some words were corrected. Shown in the manuscript in red font color, such as in line 13, etc...